



# Assessing Chinese flood protection and its social divergence

Dan Wang[1], Paolo Scussolini[2], Shiqiang Du[1,2,3,*]

[1]School of Environmental and Geographical Sciences, Shanghai Normal University, Shanghai, China

[2]Institute for Environmental Studies, Vrije Universiteit Amsterdam, Amsterdam, the Netherlands

[3]Institute of Urban Studies, Shanghai Normal University, Shanghai, China

*Correspondence to*: Shiqiang Du (shiqiangdu@shnu.edu.cn)

**Abstract.** China is one of the most flood-prone countries, and development within floodplains is intensive. However, flood protection levels (FPL) across the country are unknown, hampering the present assertive efforts on flood risk management. Based on the flood-protection prescriptions contained in the national flood policies, this paper develops a FPL dataset of China

and investigates how China should be protected accordingly and the divergent protections between demographic groups. The dataset agrees with local flood protection plans in 34 of archived 51 counties, validating the policy-based FPLs as a reliable proxy for actual FPLs. The FPLs are much higher than that in the previous global dataset, suggesting Chinese flood risk may have been overestimated. High FPLs (≥50-year return period) are seen in 282 or only 12.6% of the evaluated counties, but with a majority (55.1%) of the total exposed population. However, the low-FPL counties (<50-year return period) host a

disproportionate share (52.3%) of the exposed vulnerable population (children and elders), higher than their share (44.9%) of the exposed population. These results imply that, to reduce social vulnerability and decrease potential casualties, investment into flood risk management should also consider the demographic characteristics of the exposed population.

## 1. Introduction

Flood protection level (FPL) is the degree to which a flood-prone location is protected against flooding (Scussolini et al., 2016).

It is a key determinant of flood risk, making its quantification a prerequisite to reliable risk assessment (Ward et al., 2013). With the emergence of large-scale flood models, the necessity to quantify FPLs has increased in recent years. For example, Jongman et al. (2014) estimated the FPLs in major European river basins by assuming that high-risk areas have high FPLs. Hallegatte et al. (2013) created an FPL dataset for coastal cities through combining design information of flood defenses and



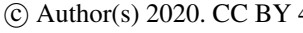

expert estimates to improve coastal flood risk assessment. Scussolini et al. (2016) developed FLOPROS, a global database of

FPLs based on information included in protection design documents and in protection policy documents, supplemented with

FPL estimates based on flood risk modeling.

Improved FPLs reduce the frequency of floods in flood-prone areas and decrease flood risk (Ward et al., 2013). From a

cost-benefit view, high FPLs are more economically attractive in areas with high density of population and economy (Ward et

al., 2017). However, high FPLs can have a 'levee effect': creating a sense of security and lowering risk awareness, which

boosts floodplain development and population growth and can in turn cause catastrophic consequences once a low-probability

flood happens (Di Baldassarre et al., 2015;Haer et al., 2020). On the other hand, low FPLs generally mean limited human and

financial resources and therefore imply a lower capacity of flood risk reduction (Cheng et al., 2018;Cross, 2001;Han et al.,

2020). Moreover, the low FPLs may coincide with a concentration of vulnerable people, e.g., the elders and children, increasing

the severity of the human consequences of floods (i.e., more likely fatalities), and more in general exacerbating the local social

vulnerability (Birkmann et al., 2016;Gu et al., 2018). Therefore, FPL study is also a key to understand the integrated socio-

hydrological system.

China is one of the countries that experience the most serious floods and the fastest urbanization. Each year between 1990

and 2017, floods in China affected 149 million people, led to 2165 deaths, and caused an economic damage of US$ 34 billion

(Du et al., 2019). Moreover, flood risk changes rapidly due to socioeconomic dynamics (Du et al., 2018) and, in the longer-

term, due to climate change (Alfieri et al., 2017;Winsemius et al., 2018). For instance, Du et al. (2018) found that urban lands

in the floodplain increased by 26,430 km$^2$, i.e., 542%, from 1992 to 2015, a process which is still in full swing and thus likely

to exacerbate flood risk in the future. Moreover, the urbanization process witnesses an enormous migration from countryside

to cities (Liu and Li, 2017;Li et al., 2018), which selectively leaves the vulnerable population behind, and may increase social

vulnerability in the countryside (Cheng et al., 2018).

However, little information is available about China's FPLs. The existing few studies of China's FPLs are only about a

specific flood control facility and at local scales. For example, Deng et al. (2015) analyzed the FPL of Taihu Lake levees to

inform flood risk management in catchments. Zhou (2018) studied the impact of land subsidence on the FPL in the downstream

of Daqing River. Liu (2017) inferred the FPL in Quzhou through hydrodynamic simulation. Although the global database

FLOPROS can show an overall FPL for China, it still misses details. For example, in the FLOPROS the FPLs are only of 20-

year return period in 29 (or 85.3%) of the 34 Chinese provinces including the capital Beijing, which is probably incompatible

with the massive Chinese investment in improving FPLs in the past decades particularly for metropolises (Du et al., 2019).

Therefore, it is reasonable to presume that the FPLs of China are significantly underestimated in the FLOPROS, especially for

urban areas. With such sparse data, the picture of FPL on the national scale is still unclear, representing a critical knowledge

gap in the context of rapid urbanization. This also limits the understanding of the relationship between population exposure,

vulnerability, and flood protection.

Therefore, the paper here develops and validates the first FPL dataset for China, based on the current Chinese policy on

FPLs. On this basis, the following questions are addressed. 1) What level of protection against river floods does Chinese policy

imply across the country? 2) Does the FPL policy take into account relevant demographics of the exposed population, such as

elders and children who are known to be most vulnerable to floods?

**2. Materials and methods**

**2.1 China's flood protection policy and the study framework**

FPL data are typically difficult to access at a large scale. Scussolini et al. (2016) proposed that FPL can be assessed based on

protection plan documents, policies, or assumed based on hydrodynamic and flood risk simulations and on wealth distribution.

Flood protection policy documents generally contain information on how a region should be protected from floods and provide

an opportunity to establish a large-scale FPL dataset (Mokrech et al., 2015;Jonkman, 2013). Presently, the key policy document

for China is the *Standard for flood control (No: GB 50201-2014)* which was released in 2014 by the *Ministry of Housing and*

*Urban-Rural Development of the People's Republic of China*. It stipulates the FPLs for urban and rural areas depending on

three exposure indicators: the amount of exposed population, the per-capita GDP (gross domestic product), and the arable

lands. Here a framework is developed using spatial information on these indicators of exposure, to infer the policy-prescribed

FPLs across China (Fig. 1).

[Insert Fig. 1]

This is conceptually akin to the *policy layer* of the FLOPROS dataset, but the framework yields information at the much finer spatial scale of the county. In the study framework, the FPL of an urban county (in Chinese: *shi* or *qu*) is evaluated by population exposure and GDP-weighted population exposure; the FPL of a rural county (*xian*) is evaluated by the population

exposure and arable-land exposure (Table 1), as prescribed in the *Standard for flood control*. Additionally, local flood protection plan documents are collected to verify the policy-based FPL dataset. The spatial pattern of the FPLs is identified using spatial statistic techniques, and the FPL of the exposed (vulnerable) population are evaluated.

[Insert Table 1]

## 2.2 Data

Six datasets are employed. First, an administrative boundary is adopted from He et al. (2016), which considered administrative boundary adjustments from 1990 to 2010. Second, a fluvial flood depth map with a 100-year return period is provided by the CIMA foundation (Rudari et al., 2015), which is accessible from the Global Risk Data Platform (http://preview.grid.unep.ch/). It has a spatial resolution of 1 km and has been used for analyzing China's urban land expansion (Du et al., 2018) and population dynamics in floodplains (Fang et al., 2018). Third, population density maps for 1990 and 2015 are acquired from

the China Temporal Dataset of Harvard Dataverse, published by the WorldPop program (http://www.worldpop.org.uk). It originally has a spatial resolution of 100 m and is aggregated to a 1 km resolution to match the flood depth data, further to get population exposure using methods described in Fang et al. (2018). Fourth, the demographic information of 1990 and 2015 is obtained from China's national census data (National Bureau of Statistics of China, 2015). It includes the proportions of children (aged $\leq$ 14 years) and elders (aged $\geq$ 65 years) to the county-level total population, which are used to calculate the

vulnerable population exposure. Fifth, the land use data of China for 2015 comes from the Data Center of Resources and Environmental Sciences, Chinese Academy of Sciences, which is provided on the Resource and Environment Data Cloud Platform (http://www.resdc.cn/). It has a resolution of 1 km and is used to extract arable lands in floodplains. Besides, the county-level gross domestic product (GDP) in 2015 is used to calculate the GDP-weighted population exposure.



### 2.3 Assessment of flood protection level

Three exposure indicators are employed to assess the FPL of a certain flood-prone county: population exposure (PopE), GDP-weighted PopE, and arable-land exposure (ArableE). For county *i*, the PopE is the population in the floodplain and is calculated by overlaying the flood depth and the population density maps using a geographical information system (Fang et al., 2018). Floodplain is defined as the maximum extent (i.e., where flood depth > 0 cm) of the 100-year flood map, which is consistent with previous flood exposure analyses (Du et al., 2018;Fang et al., 2018;Jongman et al., 2012). Then, for an urban county *i*,

the PopE is transformed into the GDP-weighted PopE using the relative factor of the county's GDP per capita to the national average GDP per capita, following Eq. (1),

$$GDP\text{-}weighted\ PopE = PopE \times \frac{G_i}{G_a} \tag{1}$$

where $G_i$ refers to the GDP per capita in county *i;* and $G_a$ refers to the national average of GDP per capita in China.

For a rural county *i*, the ArableE is calculated as the area of arable lands in the 100-year floodplain by overlaying the flood

depth and land use maps. Based on the three calculated indicators, the FPL can be estimated by applying the criteria specified in the *Standard for flood control (GB 50201-2014)* (Table 1). The FPL of an urban county is the larger value between FPLs based on the PopE criteria $FPL_{(PopE, i)}$ and based on the GDP-weighted PopE criteria $FPL_{(GDP\text{-}weighted\ PopE, i)}$, while the FPL of a rural county is the larger value between $FPL_{(PopE, i)}$ and the FPL based on the ArableE criteria $FPL_{(ArableE, i)}$, following Eq. (2),

$$FPL(i) = \begin{cases} \max[FPL_{(PopE, i)}, FPL_{(GDP\text{-}weighted\ PopE, i)}], & i = \text{an urban county} \\ \max[FPL_{(PopE, i)}, FPL_{(ArableE, i)}], & i = \text{a rural county} \end{cases} \tag{2}$$

The FPLs are assessed for 2237 counties that fully or partially fall within the 100-year floodplain. The result is presented by six FPLs: ≥ 200 years, 100–200 years, 50–100 years, 30–50 years, 20–30 years, and 10–20 years. Further, ≥50-year FPLs are summarized as relatively high FPL, while <50-year FPLs are summarized as low FPL.

### 2.4 Verification of the flood protection levels

Local documents of flood protection plans are collected for 51 counties to verify the policy-based FPL results. From those

documents, the planned FPLs are derived, which are akin to the design layer of the FLOPROS (Scussolini et al., 2016).





Assuming that the planned flood protection standards reflect the reality of flood protection implemented in practice, the agreement between the policy layer and the design layer is checked for each 51 counties and an overall accuracy is further calculated.

## 2.5 Pattern clustering of flood protection level

The LISA (local indicator of spatial association) or local Moran's *I* (Anselin, 1995) is used to identify spatial relationships between a county's FPL and its neighboring FPLs. The local Moran's *I* statistic is calculated as follows (Chakravorty et al., 2003),

$$I_i = \frac{(x_i - \bar{x})}{s^2} \sum_{j=1, j \neq i} W_{i,j} (x_j - \bar{x})$$ (3)

where $I_i$ is the local Moran's *I* in county *i*; $x_i$ and $x_j$ refer to the FPL of county *i* and its neighboring county *j*, respectively; $\bar{x}$

is the mean FPL across all counties; $W_{i,j}$ is a n-by-n weight matrix defining the spatial contiguity between county *i* and any county *j*, where $W_{i,j}$=1 if county *i* and county *j* share a border and otherwise $W_{i,j}$=0; $s^2$ is the variance of FPLs across all flood-prone counties.

A positive value for the local Moran's *I* statistic indicates that FPL in a county is similar to those in its neighboring counties, while a negative *I* value indicates dissimilar values (Zhu et al., 2018;Frigerio et al., 2018;Shen et al., 2019). The local Moran's

*I* is calculated by applying the Queen Contiguity matrix in software GeoDa (version 1.12), which is available from http://geodacenter.github.io. The significance is evaluated at an alpha level of 0.05. Four different LISA clustering patterns of FPLs are identified. 1) High-High: both the county and its neighbors have high FPL. 2) High-Low: the FPL is high in a county while low in its neighboring counties. 3) Low-High: FLP is low in a county while high in its neighboring counties. 4) Low-Low: both a county and its neighbors have low FPL.

## 2.6 Dynamic analysis of population exposure and vulnerable population exposure

*PopE* refers to the population exposure in a certain county, which is calculated as the population in the floodplain by overlaying the flood depth and the population density maps (Sect. 2.3). Exposed vulnerable population comprises the exposed children and elders because children and elders are generally considered more vulnerable to flooding, due to limited mobility and





physical resistance (Gu et al.;Salvati et al., 2018). Assuming that the proportion of exposed children to the total population is

spatially homogeneous within a county, the exposed children are calculated in each county using Eq. (4),

$$Exposed\ children = \frac{children}{total\ population} \times PopE \qquad (4)$$

where *PopE* refers to the population exposure in the county; *children* and *total population* are the number of children and total

population in a county, respectively. Similarly, the exposed elders can be calculated. The exposed vulnerable population is the

sum of exposed children and exposed elders.

Equation (5) is used to calculate the growth rate of population exposure from 1990 to 2015,

$$Growth\ rate\,(\%) = \frac{PopE_{2015} - PopE_{1990}}{PopE_{1990}} \times 100\% \qquad (5)$$

where *PopE$_{2015}$* and *PopE$_{1990}$* refer to the population exposure in 2015 and 1990, respectively. Similarly, the growth rates of

exposed children, elders, and vulnerable population are calculated.

## 3. Results

### 3.1 Validation of the new policy-based FPL dataset

The policy-based FPL dataset matches to a good degree the information from protection plan documents. In 34 (66.7%) out of

the 51 verification counties, the FPLs agree with the local official protection plans (full information in Supplement). The FPLs

in the dataset are overestimated in four counties and underestimated in five counties. Both the overestimations and

underestimations are only off by one protection level. In comparison, the FPLs of the global FLOPROS database match the

protection plan documents in 13 (25.5%) out of the 51 counties, and they underestimate FPLs in the other counties by at least

one protection level. Therefore, the policy-based FPL dataset constitutes a substantial improvement on previous knowledge of

Chinese FPLs.

### 3.2 Spatial pattern of flood protection level

According to the prescriptions of the policy *Standard for flood control*, a majority (87.4%, or 1955) of Chinese counties have

<50-year FPLs that are defined hereafter as relatively low FPLs (Fig. 2), while only 282 counties (12.6%) have high FPLs

(≥50 years). A considerable proportion (33.1%, or 741) of the evaluated Chinese counties are protected with a ≥30-year FPL,

which is much higher than that in the global FLOPROS database (Scussolini et al., 2016), in which only 5 (14.7%) out of 34

provinces have ≥30-year FPLs. Therefore, Chinese FPLs are significantly underestimated in previous studies.

[Insert Fig. 2]

The FPLs show significant divergence between eastern and western China (Fig. 2, Fig. 3), reflecting general differences in

population exposure and in economic performance. A dominant portion (85.5%, or 241) of high-FPL counties are located in

eastern China. Particularly, all the 25 counties with the highest FPL (≥200 years) are located in the east. In contrast, only 4.3%

(or 41) of the western Chinese counties have high FPLs, which is much lower than the share in eastern China (18.7%, or 241).

The majority (68.4%, or 210) of the lowest FPL counties (10–20 years) are located in the west. In sum, western China is

disproportionally protected with low FPLs.

[Insert Fig. 3]

"High-high" FPL clusters include 112 counties. They are mainly located in the three primary urban agglomerations of the

Beijing-Tianjin-Hebei, the Yangtze River Delta, and the Pearl River Delta (Fig. 4). The three primary urban agglomerations

are home to most of the counties with the highest FPLs of ≥200 years (Fig. 3). Besides, the "high-high" FPL clusters are also

located in the middle Yangtze River reaches. The "low-high" FPL clusters include a total of 66 counties, surrounding the "high-

high" FPL clusters. These counties within the "low-high" FPL clusters can be more vulnerable when they are needed to

sacrifice to protect their surrounding large cities that are more expensive to be flooded (Wang et al., 2016). "Low-low" FPL

clusters include 158 counties, which are mainly located in southwestern China and scattered along a belt from Hohhot to

Kunming. Surrounding the "low-low" FPL clusters, 48 counties have relatively high FPLs and form "high-low" clusters.

[Insert Fig. 4]

**3.3 Protection levels of the exposed (vulnerable) population in 2015**

A majority (55.1%, 231.1 million) of the total exposed population is found in a minority of 282 counties with high FPLs (≥50

years) (Table 2). Particularly, 23.3% (97.8 million) of the total exposed population are protected by a ≥200-year FPL. In





contrast, 188.4 million (44.9%) of the flood-exposed people are in low-FPL counties, lower than that in the high-FPL counties.

A majority (52.3%, 38.3 million) of the exposed vulnerable population are concentrated in low-FPL (<50 years) counties, higher than these counties' share of the total exposed population (44.9%) (Table 2). These low-FPL counties host 52.9% (19.4 million) of the exposed children and 51.6% (18.8 million) of the exposed elders. Particularly, counties with a 20–30 years FPL host the largest exposed vulnerable population (19.9 million, 27.1%) across all the six FPLs, including 10.1 million (27.5%) children and 9.8 million (26.8%) elders.

The ratio of vulnerable population to the total exposed population is as high as 20.3% in low-FPL counties, while it is 15.1% in high-FPL counties (Table 2). Both exposed children and elders are found disproportionally in the low-FPL counties. Specifically, the children's share of the total exposed population is 10.3% in the low-FPL counties, higher than in the high-FPL counties (7.5%); similarly, the elders' share is 10.0% in the low-FPL counties, higher than in the high-FPL counties (7.6%). Therefore, the low-FPL counties have a disproportionally high share of vulnerable population than the high-FPL counterparts, 195     in terms of both exposed children and elders.

[Insert Table 2]

**3.4 Changes in the exposed (vulnerable) population across protection levels**

The total exposed population has grown by 60.3%, rapidly from 1990 to 2015 in counties that are presently protected by high FPLs, while it has remained relatively stable in the low-FPL counties (2.34%) (Fig. 5a). In 1990, the exposed population was 200     primarily located in counties with 20–30 years FPLs (95.0 million, 28.9%), while in 2015 it is primarily in counties with ≥200-year FPLs (97.8 million, 23.3%).

[Insert Fig. 5]

    The exposed vulnerable population has decreased by 41.9%, from 126.0 million in 1990 to 73.3 million in 2015; and decreased more sharply (by 53.7%) in the low-FPL counties (Fig. 5b). The decrease of the exposed vulnerable population is 205     mainly caused by a sharply declining exposed population of children. The exposed children, in total, has decreased by 65.6% from 106.8 million in 1990 to 36.8 million in 2015. The exposed children's share to the total exposed population has declined rapidly across all FPLs, which decreases the ratio of vulnerable population to the total exposed from 38.4% in 1990 to 17.5%





in 2015 (Fig. 6).

In contrast, the exposed elders has increased across all the six FPLs, with a total growth by 90.2% from 19.2 million to 36.5 million (Fig. 5b). This trend reflects China's aging population. Moreover, the elders' share of the total exposed population has risen from 5.9% in 1990 to 8.7% in 2015 (Fig. 6). Particularly in the low-FPL counties, it has increased from 5.7% in 1990 to 10.0% in 2015 with a growth of 4.3%, much higher than that in the high-FPL counties (1.7%).

[Insert Fig. 6]

## 4. Discussion

### 4.1 Residual flood risk is nonstationary and should be effectively managed

Chinese FPLs should be much higher than that in previous studies, according to the prescriptions of the policy *Standard for flood control*. The newly developed data show that almost one third (33.1%, 741) of the evaluated Chinese counties are protected with a ≥30-year FPL, while this FPL is only in 5 (14.7%) out of 34 provinces in the FLOPROS (Scussolini et al., 2016). Particularly, the newly developed data show that a considerable proportion (12.6%, or 282) of Chinese counties have ≥50-year FPLs that are defined as relatively high FPLs. Moreover, the high-FPL counties protect the majority (55.1%, 231.1 million) of exposed Chinese population. Those high-FPL counties are concentrated in eastern China, particularly in the urban agglomerations (Bai et al., 2014). The underestimate of Chinese FPLs can at least partially explain the high level of flood risk in previous studies (Willner et al., 2018a;Willner et al., 2018b;Alfieri et al., 2017). For instance, global flood risk assessments show huge flood risk across Chinese provinces both in current condition and future scenarios (Willner et al., 2018a), which are considered to further propagate an devastating indirect impact to other countries through the global trade and supply network (Willner et al., 2018b). However, those global assessments are based on the FLOPROS database, which significantly underestimate Chinese FPLs, e.g., presenting Beijing with a 20-year FPL, which should be 200 years in the newly developed result (Fig. 3) and in the local official document (full information in Supplement). The real flood risk should thus be much lower than the estimates in previously studies if the new FPL is considered.

However, the high flood protection does not represent absolute safety. On the contrary, low-probability floods can still





occur and flood protection structures may technically fail, causing residual flood risk (Haer et al., 2020). Particularly, levee breaches can cause a catastrophe for the areas with high density of population and assets (Jongman, 2018). In high-FPL counties, a sense of safety brought by the flood protection structures can reduce the perception of risk and cause "levee effect" — boosting floodplain development and increasing flood exposure (Cheng and Li, 2015;Kates et al., 2006). Such a phenomenon is probably at play in China, as suggested by the faster increase of the exposed population in the high-FPL counties than in the low-FPL counties. The rapid increase in the exposed population can exacerbate residual flood risk, rendering these high-FPL areas vulnerable to low-probability and high-impact floods (Koks et al., 2015;Di Baldassarre et al., 2013). The residual risk can be further aggravated by future climate change (Alfieri et al., 2017;Winsemius et al., 2018). Alfieri et al. (2017) indicated that the future annual expected economic losses in China may be the highest of all countries, rising by 1.5-fold to 3.4-fold and reaching 50 to 110 billion EUR/year, based on global warming scenarios of 1.5℃ to 4℃, respectively.

The residual flood risk can be higher if the real-world flood protection lags behind the policy requirement and design. In fact, the policy-based FPL dataset only reflects how a county should be protected according to the flood protection policy, which also stipulates that the flood protection should be updated along with population growth and economic development. Unfortunately, a survey of 2013 found that 44% (284) of the 642 Chinese cities did not update their flood protection planning according to their socioeconomic growth (Cheng and Li, 2015). A neglect of the real-world flood protection lagging behind the policy-based flood protection can distort the selection of adaptation measures.

Flood risk management will inevitably face an ongoing challenge from population growth when reducing the residual risk. It is predicted that, in the next ten years, Chinese urban population will increase by 17% (United Nations, 2018), which will increase residual flood risk in the high-FPL counties because high-FPL counties are usually urban areas. The flood protection structures should be upgraded along with socioeconomic development and climate change to keep the residual flood risk to an acceptable level (Kwadijk et al., 2010). Non-structural measures such as early warning systems, land use planning, building codes, and insurance/reinsurance can be a complement to the flood protection structures for effectively managing flood risk (Aerts et al., 2014;Jongman, 2018;Du et al., 2020).



## 4.2 Demographics should be included in the flood protection policy

Although the low-FPL counties see less exposed population, the majority (52.3%, 38 million) of exposed vulnerable population are concentrated there. Particularly, the elders' share of the total exposed population was increasing rapidly in these low-FPL counties. These findings are consistent with other studies (Cheng et al., 2018). The low-FPL counties are often located in rural areas with economic downturn and insufficient job opportunities, which causes a large number of young adults to temporarily migrate to cities for work opportunities (He et al., 2016;Meng, 2014). Therefore, it may be difficult for these low-FPL counties

to respond to and recover from flooding due to economic backwardness and labor shortages.

Hence with more vulnerable people, a higher potential casualty or injury rate caused by floods is expected in these low-FPL counties. The elder Chinese are predicted to more than double from 128 million in 2015 to 348 million in 2050 (The World Bank, 2019), implying an increase of exposed elders. This will further increase social vulnerability and challenge flood risk management, particularly in the low-FPL counties. However, the policy *Standard for flood control* neglects the

demographic characteristics of the exposed population. It is economically reasonable to employ a relative low FPL for areas that have low density of population and economy. Such a strategy, however, may aggregate flood risk because the less protected areas coincide with high social vulnerability that is caused by a disproportional distribution of vulnerable people, particularly elders.

Therefore, local demographic characteristics should be considered for an economically and socially beneficial strategy of

flood adaptation (Koks et al., 2015). The low-FPL areas can employ decentralized and soft adaptation measures, such as elevation of buildings, wet flood-proofing and dry flood-proofing to reduce flood vulnerability (Aerts, 2018;Du et al., 2020), since structural flood protections are generally less cost-effective in areas with fewer exposed population (Ward et al., 2017;Jongman, 2018). Considering the relative concentration of exposed vulnerable population in the low-FPL counties, flood risk information, adaptation measures, and emergency plans should be made accessible and understandable to children and

elders (De Boer et al., 2014). Communities should pay more attention to children and elders during early warning, evacuation and resettlement; and a one-on-one assistance scheme can be developed at the community level to help the vulnerable people. Emergency plan and flood adaptation design should consider the particular needs of children and elders, which can be promoted by their participation in the planning and designing processes (Liang et al., 2017).

### 4.3 Limitations and future perspectives

The newly developed FPL dataset reflects how China should be protected against river floods according to the flood protection policy. It does not report actual FPLs although it agrees with local flood protection plans very well. Given the scarcity of the real-world flood protection data, the new dataset can be considered a valid proxy of actual FPLs, and can assist efforts to understand, evaluate, and manage flood risk. Moreover, the real-world FPLs are not fixed but are plausibly updated along with socioeconomic development and climate change. For these reasons, we invite relevant departments, communities, and users

to use, verify, and improve the newly developed FPL result (which are accessible as Supplement to this paper). With a wide participation of public stakeholders, the flood protection data can be much improved in the future.

Limitations also come with our data and methods. The exposed population is calculated based on a gridded population dataset from the WorldPop program, which is a disaggregation result of census population using auxiliary variables such as land use conditions and nightlight brightness. However, neither the disaggregation methods nor the auxiliary data are free of

uncertainty and error (Smith et al., 2019). Moreover, due to a lack of gridded demographic data, the exposed vulnerable population is calculated assuming that its proportion to the total population is spatially homogeneous within a county. But in fact, the demographic characteristics can be spatially heterogeneous (Han et al., 2007;Qiang, 2019). Nowadays, crowd-sourcing population data are emerging, thanks to social media (Goodchild and Glennon, 2010;Smith et al., 2019) and mobile phone records (Wu et al., 2012). These new data can help to improve the exposed (vulnerable) population accuracy and in turn

the FPL estimates.

### 5. Conclusions

A framework is developed to assess the county-level FPLs in China based on the flood protection policy and relevant socioeconomic variables of floodplains. The produced FPL dataset shows a match rate of 66.7% with the planned FPLs included in specific flood protection documents in a sample of 51 counties. The policy-based FPL dataset constitutes a

substantial improvement on previous knowledge and the dataset is relatively accurate. This study thus agrees with the argument of Scussolini et al. (2016) that flood protection policy is a valid proxy for actual FPLs. However, there may be still significant



differences between the policy-based FPLs and the actual flood protection because the latter *may* be behind or ahead the policy-required FPLs. The FPL dataset was thus made open access to encourage relevant users to check and improve it.

The produced FPL dataset shows that western China is dominated by low FPLs while high-FPL counties are concentrated in the east. There are 282 counties with a high FPL (≥50 years), which account for only 12.6% of the total flood-prone counties but host 55.1% (231.1 million) of the total exposed population. In contrast, more exposed vulnerable population (52.3%, 38 million) are concentrated in the low-FPL counties. Moreover, exposed population grows rapidly (by 60.3%) in the high-FPL counties while the proportion of elders increases more rapidly in the low-FPL counties than in the high-FPL counties. These findings imply that the flood protection policy has a relatively efficient strategy to protect the majority of the exposed population within a minority of well-protected counties. However, the rapid growth of exposed population can increase residual flood risk. Moreover, the disproportional concentration and rapid increase of exposed vulnerable population, particularly the elders, in the low-FPL counties can probably increase the places' vulnerability.

Therefore, diversified adaptation measures including both structural flood defenses and non-structural solutions should be employed to reduce flood risk in both the high- and low-FPL counties. Local demographic characteristics should be considered for an economically and socially beneficial strategy of flood adaptation. Particularly, the vulnerable population in the low-FPL counties should receive dedicated attention. This study shows that combining FPL and demographic information is critical to understand and manage flood risk.

*Data availability.* The Chinese flood protection data are available as supplement. Supporting data are accessible through the associated references.

*Author contribution.* Shiqiang Du designed this study. Dan Wang implemented the data processing and analysis. Dan Wang, Paolo Scussolini and Shiqiang Du prepared the manuscript with contributions from all co-authors.

*Competing interests.* The authors declare that they have no conflict of interest.

*Acknowledgements.* This research was funded by the National Natural Science Foundation of China (Grant Nos. 41871200, 41730642, and 51761135024), the National Key Research and Development Program of China (2017YFC1503001), and the Netherlands Organization for Scientific Research (NWO, grant no. ALWOP.164).



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





**Table 1.** Urban and rural standards for evaluating the flood protection levels (FPL) (source: Standard for flood control GB 50201-2014)

| Urban FPL Indicators | | | Rural FPL Indicators | | |
|---|---|---|---|---|---|
| Population exposure (million) | GDP-weighted population exposure* (million) | Urban FPL (Return period, years) | Population exposure (million) | Arable lands exposure (thousand ha) | Rural FPL (Return period, years) |
| ≥1.5 | ≥3 | ≥200 | ≥1.5 | ≥200 | 50–100 |
| ≥0.5 | ≥1 | 100–200 | ≥0.5 | ≥66.7 | 30–50 |
| ≥0.2 | ≥0.4 | 50–100 | ≥0.2 | ≥20 | 20–30 |
| <0.2 | <0.4 | 30–50 | <0.2 | <20 | 10–20 |

Note: *population exposure is multiplied by relative per capita gross domestic product (GDP) to the national average.





**Table 2.** Exposed population (total, vulnerable, children and elders) and vulnerable population, in absolute amounts and as percentage of the total population, in counties with different flood-protection-level in 2015.

| FPL (years) | Total exposure in millions (%) | Vulnerable exposure in millions (%) | Exposed children in millions (%) | Exposed elders in millions (%) | Vulnerable population ratio* (%) |
|---|---|---|---|---|---|
| **High** | **231.1 (55.1)** | **35.0 (47.7)** | **17.3 (47.1)** | **17.7 (48.4)** | **15.1** |
| ⩾200 | 97.8 (23.3) | 12.4 (17.0) | 5.9 (16.2) | 6.5 (17.8) | 12.7 |
| 100–200 | 82.5 (19.7) | 13.6 (18.6) | 6.9 (18.8) | 6.7 (18.3) | 16.5 |
| 50–100 | 50.8 (12.1) | 9.0 (12.2) | 4.5 (12.1) | 4.5 (12.3) | 17.6 |
| **Low** | **188.4 (44.9)** | **38.3 (52.3)** | **19.4 (52.9)** | **18.8 (51.6)** | **20.3** |
| 30–50 | 89.9 (21.4) | 18.0 (24.6) | 9.2 (24.9) | 8.9 (24.3) | 20.0 |
| 20–30 | 96.5 (23.0) | 19.9 (27.1) | 10.1 (27.5) | 9.8 (26.8) | 20.6 |
| 10–20 | 2.0 (0.5) | 0.4 (0.6) | 0.2 (0.6) | 0.2 (0.6) | 21.0 |
| Sum | 419.5 (100) | 73.3 (100) | 36.8 (100) | 36.5 (100) | 17.5 |

Note: * refers to the share of vulnerable people to total exposed population in a region.


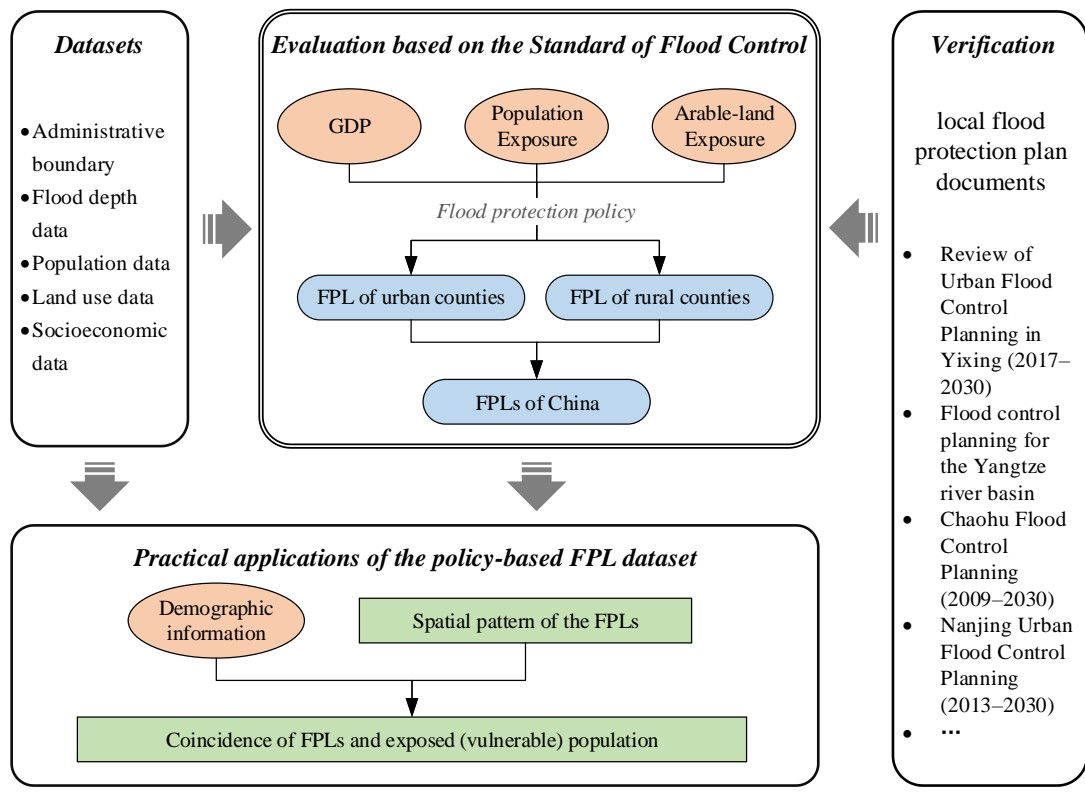

**Figure 1** The study framework of flood protection level (FPL). Input datasets are indicated in orange circles; the new datasets produced in this study are indicated in blue rounded rectangle; the implications of the new datasets are in green rectangles.




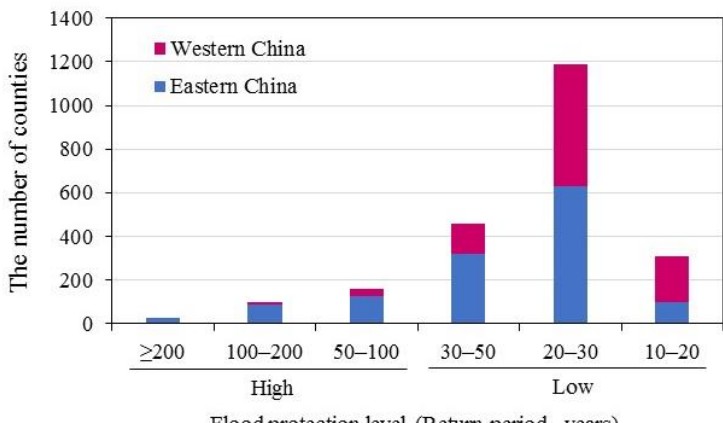

**Figure 2** The county numbers of different flood-protection levels between western and eastern China. (The boundary between western and eastern China is shown in Fig. 3)

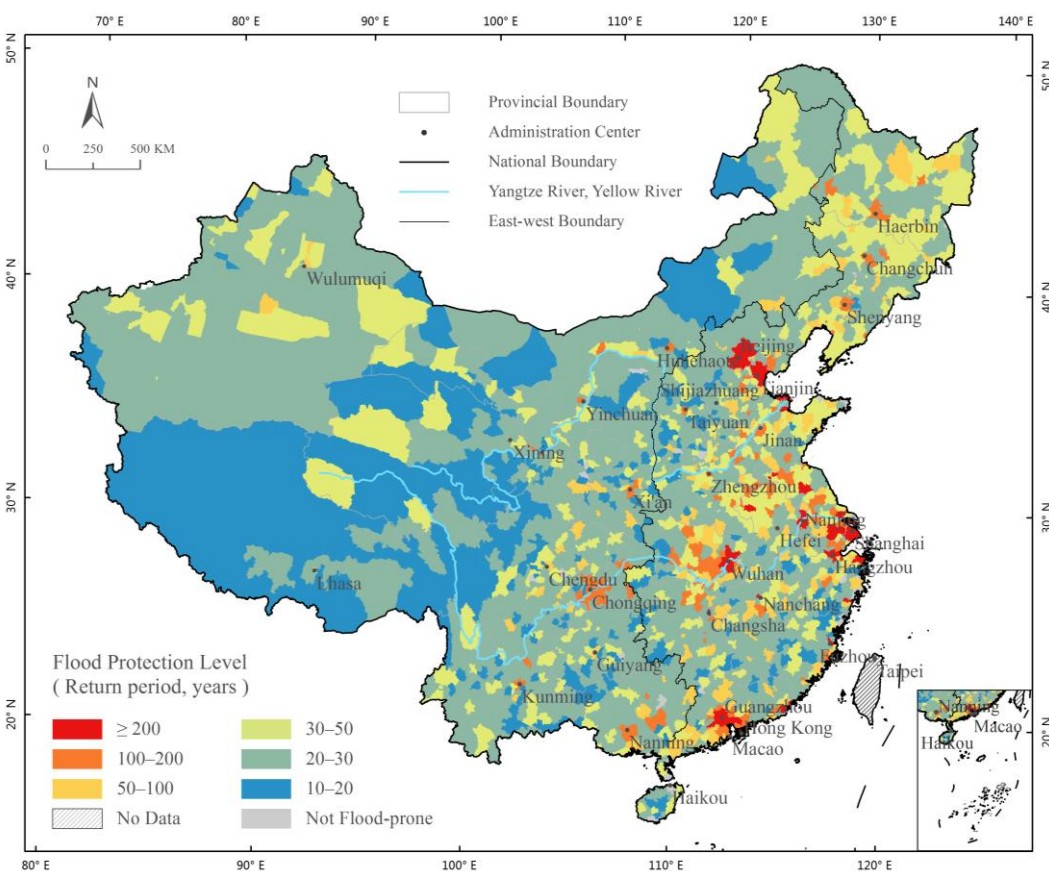

**Figure 3** Flood protection level (FPL) for each county in China. The FPL is limited to the scope of floodplains but plotted to cover the entire counties. The data should only be viewed as a valid proxy of the actual FPLs, not equating the actual FPLs. The Shapefile format data are available as Supplement.


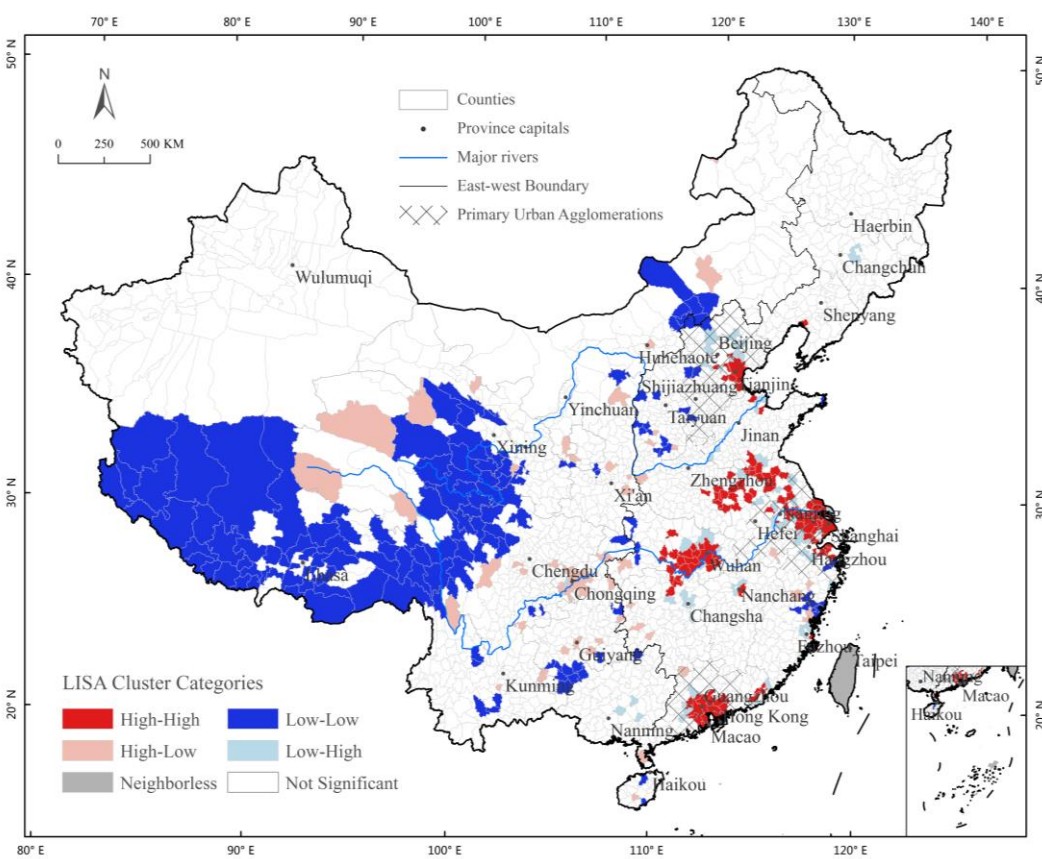

**Figure 4** Spatial cluster of flood protection level in China




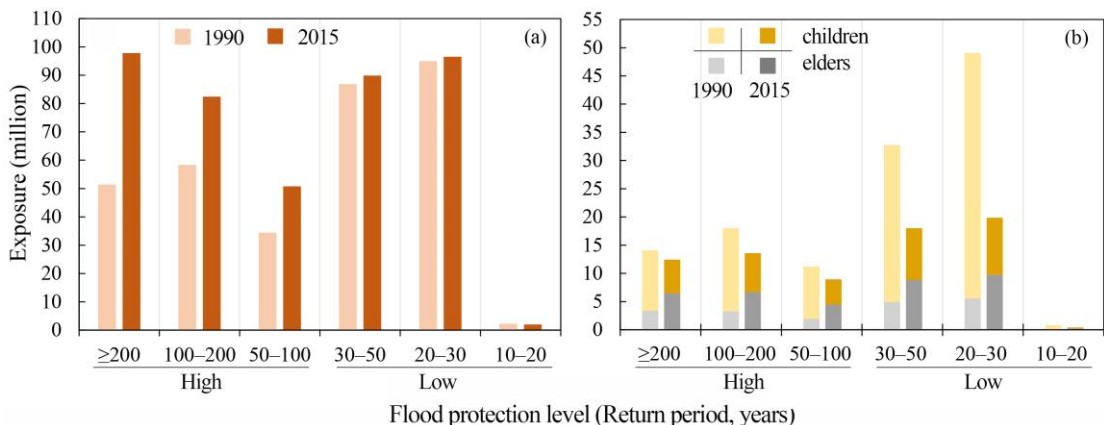

**Figure 5** Changes in exposed total population **(a)** and vulnerable population **(b)** across different flood protection levels from 1990 to 2015.
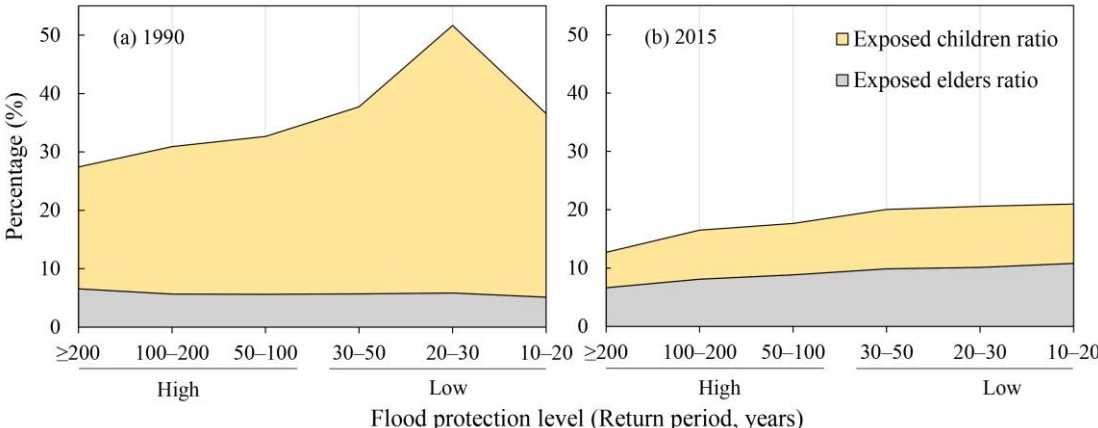

**Figure 6** The ratios of exposed vulnerable population, exposed children, and exposed elders to the total exposed population across different flood protection levels in 1990 **(a)** and 2015 **(b)**. Note that flood protection levels refer to the situation of 2015 and keep constant between 1990 and 2015.