# Peer review of "Assessing Chinese flood protection and its social divergence"

_Natural Hazards and Earth System Sciences, 2020_

## Referee Comment (RC1) · Anonymous Referee #1 · 16 Oct 2020

The manuscript developed a new food protection dataset for China based on the relevant policy and multi-source data. Such a dataset is urgently needed as it is a foundation of reliable flood risk assessment and effective risk management, but scarce in realty. This dataset revealed how an area should be protected according to the relevant policy. Thus it helped to identify the potential social divergence and the vulnerable groups in terms of lower flood protection. There is a limited amount in the literature on this topic, so it fills an important gap. The manuscript is generally well written and interesting. Specific comments are as follows. 1. lines 12-13. The validation can only reveal that the policy-based FPLs is a reliable proxy for the actual FPLs in Chinese case. It should be with caution to extend the conclusion. 2. lines 13-14. More explanations are needed on how Chinese flood risk may have been overestimated. 3. lines

62, references are needed to say the FPL data are not well accessible. 4. lines 93, the data source of the GDP data should be specified. 5. Table 2, the caption is unclear. Are the vulnerable exposed population in the brackets different from the followed vulnerable population? 6. figure 2, the axis of flood protection levels should increase from the left to the right. 7. Figure 3, the boundary lines are difficult to identify, particularly for the provincial level.
* * *

---

## Referee Comment (RC2) · Anonymous Referee #2 · 16 Oct 2020

This paper develops a county-level flood protection level (FPL) dataset for China on the basis of the prescriptions of the 2014 Chinese policy document Standard for flood control. It then analyses this against the amount of children and elders in the country, also by county. The paper is generally well written, and even though I do not think it can be considered particularly substantial as a research article (as the bulk of the work consists in essentially overlapping GIS datasets following a policy document), I believe the results are nevertheless interesting and useful to the community. Thus, in my opinion the article may be considered for publication in NHESS, although several improvements are necessary.

General comments

1. The paper defines floodplain as the maximum extent of the 100-year flood map, and

the exposed elements are then defined as the elements within that area. Should this refer to a defended or an undefended 100-year flood extent? Is there a reference to this aspect in the policy document, i.e. a guideline on how the actual quantification of exposed elements should be performed? This point needs to be clearly addressed in the article, as it will necessarily affect the estimation of FPLs. Moreover, taking into account that the 100-year hazard map used in this study (Rudari et al., 2015) already implicitly considers the existence of flood defences based on GDP, does its application impact the estimation of FPLs? Please discuss.

2. The validation of FPLs is carried out not against a sample of actual flood protection infrastructure, but rather against local flood protection plans. Therefore, this exercise can be viewed more as a check on whether county-level flood protection policies are aligned with the national one from 2014, rather than an actual validation of computed FPLs. Although the authors acknowledge this limitation in the article, I am not convinced with statements such as "validating the policy-based FPLs as a reliable proxy for actual FPLs", which I find partly unsupported. I think the article would benefit significantly from a more robust validation with ground-truth data for a number of counties. Is this information for some counties not available or obtainable at all, e.g. with river basement management authorities?

3. Still related to the comparison of county level plans and the national policy regarding protection level, can you please provide some additional information on how these counties were selected? It would be relevant to understand if these counties are representative of the different realities in China, particularly in terms of the variables defined in the policy (rural/urban, exposed population, arable land). You found an agreement in FPLs in 66.7% of the counties – can this be attributed in some way to specific properties of these counties, for example? Additional information on the validation counties and additional discussion on this would be useful.

Specific comments

Title: I feel that the use of "social divergence" raises a reader's expectations above what is actually presented in the article, which is limited to age groups. Please adjust the title to reflect this, or otherwise expand the analysis to include other factors that influence social vulnerability – the latter would certainly be more insightful and make the article more interesting.

L37: Remove 'Each year' (I assume these are aggregate numbers for 1990-2017)

L58: I do not fully understand what the second research question means, in the sense that the policy document does not make reference to demographics in the definition of FPLs, and so the answer to this is already known. Please clarify.

L65: My interpretation of Jonkman, 2013 is that it states the actual opposite of what you are saying in this sentence. For example, Jonkman, 2013 says that "... the actual protection levels could differ by more than a factor of 10 from the protection standard, and the effect on risk will be similar." Please discuss and revise.

Eq. 1: I find "GDP-weighted PopE" a poor name for a variable, as it is a bit long and at first sight it appears to be GDP minus... Please improve.

L119: Section 2.5 is unexpected and feels disconnected from what comes before in the article, because up to this point you have not yet stated that this is an analysis you will be doing. Is this cluster analysis meant to address a research question? Please contextualize beforehand, and when doing so provide an explanation on why this analysis is useful.

L163: Unclear which previous studies this sentence refers to. Is it only Scussolini et al., 2016? Please clarify.

L197: Because FPLs also change over time but only current FPLs are considered in this section, I am unsure about the usefulness of the analysis carried out here. For the same reason, I also find this section title a bit misleading. Please improve and clarify.

L235: This could also simply be the result of FPLs being calculated on the basis of

present-time exposed population, couldn't it? We do not have information about FPLs in 1990; therefore, stating that a faster increase in exposed population may have occurred in these counties because in the past their FPL was already high seems speculative. Please discuss.

Table 1: Note at the bottom is unclear.

Figure 2: Remove "the" in y-axis label.

Figure 5: In the y-axis label, replace "Exposure" with "Exposed population".

---

## Referee Comment (RC3) · Anonymous Referee #3 · 19 Oct 2020

Journal: NHESS
Title: **Assessing Chinese flood protection and its social divergence**
Author(s): Wang et al.
MS No.: NHESS-2020-264
MS Type: Research Article
**Iteration: First review**

The objective of the paper is to develop and validates a Flood Protection Level (FPL) dataset for China, which is based on current Chinese policy on FPLs. Accordingly, base data and methodologies for its development are first discussed, and then results are critically analysed.

Although the paper does not represent any significant improvement in research, it supplies relevant information for flood risk management in one of the biggest and most flood prone area of the world; and thus, it can be of interest for the journal audience.

The paper is generally well written and organised; data, methods and results are quite well explained. However, before results can be published, shared and made available to the research community, I think that some conceptual aspects deserve more attention and clarification.

**General comments**

1.      The FPL generated by the dataset is a theoretical one (i.e. designed based) and not the real one. This must be very clear since the beginning of the paper and not marginally discussed at the end. Accordingly, authors should stress since the beginning why this information is useful, how it can be used for risk management, e.g. as a proxy of the flood risk in an area?

2.      With respect to the last point, the second research question could then be changed in: Is FPL representative of the real risk in the area or its definition/evaluation should be changed? In fact, the present second research question (i.e. does the FPL policy take into account relevant demographics of the exposed population, such as elders and children who are known to be most vulnerable to floods?) is not clear at this point of the paper (i.e. why exactly this question?) as it is too much linked to an evidence that comes out only at the end of the manuscript

3.      The validation process is very weak, so I do not agree with authors that theoretical FPL agrees with real one very well (see section 4.3). The validation process was carried out only for 51 (about 2%) out of 2237 counties and a match was observed only for 66.7% of the counties (abut 1,5%). This has important implication on the use of results (see comment 1)

4.      The calculation of FPL is based on the assumption that the exposed area coincides with the 100 years return period flooded area. As this critically affects the estimation of FPL, authors should explain the reasons of this assumption. Moreover, how such an area was derived? does the modelling consider or not the existence of flood protections? What this implies?

**Minor comments**

**Section 1: Introduction**

Pg. 1 line 21 "With the emergence of large-scale flood models, the necessity to quantify FPLs has increased in recent years" → the cause-effect relation is not clear to me, could authors comment more on this?

Pg. 1 line 27 → what "improved FPLs" means?

Pg. 2 line 37 "China is one of the countries that experience the most serious floods and the fastest urbanization. Each year between 1990 and 2017, floods in China affected 149 million people, led to 2165 deaths, and caused an economic damage of US$ 34 billion" → I guess these figures refer to average data

**Section 2**

Pg. 4 line 85 "It originally has a spatial resolution of 100 m and is aggregated to a 1 km resolution to match the flood depth data, further to get population exposure using methods described in Fang et al. (2018)" → I think that a brief explanation/recall of how the data were elaborated is required.

**Section 3**

Pg. 7 line 151 "In 34 (66.7%) out of the 51 verification counties, the FPLs agree with the local official protection plans (full information in Supplement). The FPLs in the dataset are overestimated in four counties and underestimated in five counties" → what about the other 8 counties?

Pg. 8 line 176 "These counties within the "low-high" FPL clusters can be more vulnerable when they are needed to sacrifice to protect their surrounding large cities that are more expensive to be flooded" → not clear, more vulnerable than what? Could authors explain?

**Section 4**

Pg. 10 line 217 "The newly developed data show that almost one third (33.1%, 741) of the evaluated Chinese counties are protected with a ≥30-year FPL" → should be protected…. It's a theoretical FPL

Pg. 10 line 224 "For instance, global flood risk assessments show huge flood risk across Chinese provinces both in current condition and future scenarios (Willner et al., 2018a), which are considered to further propagate a devastating indirect impact to other countries through the global trade and supply network (Willner et al., 2018b). However, those global assessments are based on the FLOPROS database, which significantly underestimate Chinese FPLs, e.g., presenting Beijing with a 20-year FPL, which should be 200 years in the newly developed result (Fig. 3) and in the local official document (full information in Supplement). The real flood risk should thus be much lower than the estimates in previously studies if the new FPL is considered" → The authors cannot made this statement as the correspondence between theoretical and real FPLs have been evaluated only for 51 out of 2237 counties; the case of Beijing is a fortunate one where a perfect match occurs. But, can authors exclude that counties exist where there is not a FPL at all in practice, in front of a theoretical FPL, or a real FPL that is lower to designed based one? In this case, the risk can be underestimated. Please, comment.

Pg. 11 line 245 "A neglect of the real-world flood protection lagging behind the policy-based flood protection can distort the selection of adaptation measures" → this is exactly the point. Then, how theoretical FPL can be used (see general comment 1)?

Pg. 12 line 266 "Such a strategy, however, may aggregate flood risk because the less protected areas coincide with high social vulnerability that is caused by a disproportional distribution of vulnerable people, particularly elders" → what authors mean with "aggregate flood risk"

**Section 5**

Pg. 13 line 300 "This study thus agrees with the argument 300 of Scussolini et al. (2016) that flood protection policy is a valid proxy for actual FPL" → I do not agree, see general comment 4

**Table and Figures**

Figure 1 → I think that a full description of the framework is required in the text, i.e. in Section 2.1, to support readers in the full comprehension of following contents.

Figure 3 → colours used for the FPLs 30-50 and 50-100 cannot be distinguished in the figure.

---

## Author Comment (AC1) · 30 Nov 2020

**Response to the Anonymous Referee #1**

***General Comments.*** The manuscript developed a new food protection dataset for China based on the relevant policy and multi-source data. Such a dataset is urgently needed as it is a foundation of reliable flood risk assessment and effective risk management, but scarce in realty. This dataset revealed how an area should be protected according to the relevant policy. Thus it helped to identify the potential social divergence and the vulnerable groups in terms of lower flood protection. There is a limited amount in the literature on this topic, so it fills an important gap. The manuscript is generally well written and interesting. Specific comments are as follows.

*Accepted*:  Thanks for confirming the relevance of our manuscript and the suggestions for further improvement. We have thoroughly revised our paper, addressing your valuable comments and suggestions.

***Specific Comment 1***.  Lines 12-13. The validation can only reveal that the policy-based FPLs is a reliable proxy for the actual FPLs in Chinese case. It should be with caution to extend the conclusion.

*Accepted*:  Thanks for this suggestion. We have revised the sentence. Please check from *line 12 on page 1*. Now it reads as:

> This suggests that the policy-based FPLs is a valuable proxy for actual FPLs in China.

***Specific Comment 2***.  Lines 13-14. More explanations are needed on how Chinese flood risk may have been overestimated.

*Accepted*:  Thanks for the suggestion. The overestimation of Chinese flood risk in previous studies resulted from an underestimation of Chinese flood protection. We revised the sentence accordingly. Please check from *lines 12–14 on page 1* or as follows:

> The FPLs are significantly higher than previously estimated in the FLOPROS global dataset, suggesting that Chinese flood risk was probably overestimated.

Further, we compared the FPL dataset against the FLOPROS using the Paired Sample T Test and found that the protection levels are significantly higher in the former than in the latter (p<0.01). Please check from *Supplementary Table S4*.

***Specific Comment 3***.  Line 62, references are needed to say the FPL data are not well accessible.

*Accepted*:  Thanks for the suggestion. References have been added *(line 66 on page 3)* and the sentence now reads as:

> FPL data are typically difficult to access at a large scale in China (Jiang et al. 2020).

*References:*

*Jiang, Y., Zhi, Y., Zhao, H., Liang, L., Cao, y., and Gu, J.: Research status and prospects on water conservancy big data, Journal of Hydroelectric Engineering, 39, 1-32, 2020.*

***Specific Comment 4***. Lines 93, the data source of the GDP data should be specified.

*Accepted*: Thanks for the suggestion. The data source was the Statistical Yearbook of Chinese Cities 2016, which has been added *(lines 100–101 on page 5)*.

*References:*

*Division of Urban Social and Economic Survey of National Bureau of Statistics: Statistical Yearbook of Chinese Cities, China Statistical Press, Beijing, 2016.*

***Specific Comment 5***. Table 2, the caption is unclear. Are the vulnerable exposed population in the brackets different from the followed vulnerable population?

*Accepted:* Thanks for the suggestion. The caption of Table 2 has been clarified. Please check from *lines 519–521 on page 22*, or as below.

**Table 2.** Exposed population (total, vulnerable, children, and elders) for each flood protection level (FPL), in absolute amounts and as percentage of the whole exposed population. The rightmost column reports the ratio of vulnerable to the total exposed population.

| FPL (years) | Total exposure in millions (%) | Vulnerable exposure in millions (%) | Exposed children in millions (%) | Exposed elders in millions (%) | Vulnerable-to-total exposed population ratio |
|---|---|---|---|---|---|
| **Low** | **188.4 (44.9)** | **38.3 (52.3)** | **19.4 (52.9)** | **18.8 (51.6)** | **20.3%** |
| 10–20 | 2.0 (0.5) | 0.4 (0.6) | 0.2 (0.6) | 0.2 (0.6) | 21.0% |
| 20–30 | 96.5 (23.0) | 19.9 (27.1) | 10.1 (27.5) | 9.8 (26.8) | 20.6% |
| 30–50 | 89.9 (21.4) | 18.0 (24.6) | 9.2 (24.9) | 8.9 (24.3) | 20.0% |
| **High** | **231.1 (55.1)** | **35.0 (47.7)** | **17.3 (47.1)** | **17.7 (48.4)** | **15.1%** |
| 50–100 | 50.8 (12.1) | 9.0 (12.2) | 4.5 (12.1) | 4.5 (12.3) | 17.6% |
| 100–200 | 82.5 (19.7) | 13.6 (18.6) | 6.9 (18.8) | 6.7 (18.3) | 16.5% |
| ≥200 | 97.8 (23.3) | 12.4 (17.0) | 5.9 (16.2) | 6.5 (17.8) | 12.7% |
| **Sum** | **419.5 (100)** | **73.3 (100)** | **36.8 (100)** | **36.5 (100)** | **17.5%** |

***Specific Comment 6***. Figure 2, the axis of flood protection levels should increase from the left to the right.

*Accepted:* Thanks for the suggestion. Figure 2 has been clarified. Please check from *lines 526–527 on page 24*, or as below.

[Figure]

**Figure 2** The number of counties with different flood protection levels. (The map of western and eastern China is shown in Figure 3)

***Specific Comment 7***.    Figure 3, the boundary lines are difficult to identify, particularly for the provincial level.

***Accepted:***    Thanks for the suggestion. Figure 3 has been clarified accordingly. Please check from *lines 528–531 on page 25*, or as below.

[Figure]

**Figure 3** Flood protection level (FPL) for Chinese counties. The FPL is limited to the scope of floodplains but plotted to cover the entire counties. The data should only be viewed as a proxy of the actual FPLs, not equating to the actual FPLs. The Shapefile format data are available as a supplement.

---

## Author Comment (AC2) · 30 Nov 2020

**Response to the Anonymous Referee #2**

***General Comments.*** This paper develops a county-level flood protection level (FPL) dataset for China on the basis of the prescriptions of the 2014 Chinese policy document Standard for flood control. It then analyses this against the amount of children and elders in the country, also by county. The paper is generally well written, and even though I do not think it can be considered particularly substantial as a research article (as the bulk of the work consists in essentially overlapping GIS datasets following a policy document), I believe the results are nevertheless interesting and useful to the community. Thus, in my opinion the article may be considered for publication in NHESS, although several improvements are necessary.

*Accepted*:   Thanks for confirming the relevance of our manuscript and the suggestions for further improvement. We have thoroughly revised the paper, addressing your valuable comments and suggestions.

***General comment 1.***   The paper defines floodplain as the maximum extent of the 100-year flood map, and the exposed elements are then defined as the elements within that area. Should this refer to a defended or an undefended 100-year flood extent? Is there a reference to this aspect in the policy document, i.e. a guideline on how the actual quantification of exposed elements should be performed? This point needs to be clearly addressed in the article, as it will necessarily affect the estimation of FPLs. Moreover, taking into account that the 100-year hazard map used in this study (Rudari et al., 2015) already implicitly considers the existence of flood defences based on GDP, does its application impact the estimation of FPLs? Please discuss.

*Accepted*:   Thank you for the suggestion. The 100-year flood map we applied, which was provided by Dr. Roberto Rudari from the CIMA Foundation, is explicitly based on undefended terrain. The undefended data were used instead of the defended one for two major reasons. First, the flood defenses were generally designed based on the potentially protected population and assets that would be under threat of floods prior to the defenses. Therefore, the exposed elements should be identified from the undefended flood maps, which provides a clue for inferring the flood defenses, as shown in the Chinese flood control policy. Second, flood defenses cannot ensure the protected areas' absolute safety; thus, the population and assets should not be excluded from flood exposure analysis. We now specify this important feature in the manuscript, also following General Comment 4 of Referee #3. Please check from *lines 88–91 on page 4*.

Further, we clarified how we defined the flood exposure, also following General Comment 4 of Referee #3. The flood exposure was calculated as the elements within the maximum extent of the 100-year return period flood. This definition is consistent with the flood risk assessment by Shi et al (2015) and the flood exposure analysis by Jongman et al (2012), Du et al (2018), and Fang et al (2018). Please check from *lines 104–107 on page 5*.

*References:*

*Du S, He C, Huang Q, Shi P, 2018. How did the urban land in floodplains distribute and expand in China from 1992–2015? Environmental Research Letters, 13(3): 034018.*

*Fang Y, Du S, Scussolini P, Wen J, He C, Huang Q, et al., 2018. Rapid Population Growth in Chinese Floodplains from 1990 to 2015. International Journal of Environmental Research and Public Health, 15(8): 1602.*

*Jongman B, Ward P J and Aerts J C J H 2012 Global exposure to river and coastal flooding: long term trends and changes Glob. Environ. Change 22 823–35*

*Shi PJ, Wang JA, Xu W, Ye T, Yang SN, Liu LY, Fang WH, Liu K, Li N and Wang M. 2015 World Atlas of Natural Disaster Risk (Heidelberg: Springer)*

***General comment 2.*** The validation of FPLs is carried out not against a sample of actual flood protection infrastructure, but rather against local flood protection plans. Therefore, this exercise can be viewed more as a check on whether county-level flood protection policies are aligned with the national one from 2014, rather than an actual validation of computed FPLs. Although the authors acknowledge this limitation in the article, I am not convinced with statements such as "validating the policy-based FPLs as a reliable proxy for actual FPLs", which I find partly unsupported. I think the article would benefit significantly from a more robust validation with ground-truth data for a number of counties. Is this information for some counties not available or obtainable at all, e.g. with river basement management authorities?

*Accepted*: Thank you for the suggestion. Indeed, the only data we can find for the validation are documents of flood protection design, rather than the actual protection due to the lack of accessible ground-truth data. We agree that the flood protections in design documents are different from actual protection. However, we believe it is plausible to assume the actual protection of protection infrastructures that are completed and qualified to be equal to or higher than the designed standards, as a result of strict and tight control in the authoritarian administration of China. Following your critical comment and General Comment 3 of Referee #3, we dedicated additional efforts to enhance the validation, increasing the validation sample size from 51 counties to 171 counties. Now, as we specify in Section 2.4, the validation samples represent 7.6% of the surveyed Chinese counties, 34.0% of the exposed population, and 13.0% of Chinese exposed arable lands. We believe the expanded sample provides a substantially more solid base for the validation. Please check from *lines 129–131 on page 6* and *Supplementary* Table S1.

Besides, we have refined the selection process in the manuscript *(lines 121–129 on page 6)*. We selected the protection design documents for a relatively recent period from 2007 to 2012, neither too old that may be outdated nor too new that may be uncompleted and unqualified. Those documents would be kept in the validation data only if they stated that the design would be completed by 2015. Additionally, new flood protection design documents starting from 2015 were also employed, only if they stated the current (2015) flood protection standards.

Additionally, we revised the sentence about the validation statement. Now, it reads as: "This suggests that the policy-based FPLs is a valuable proxy for actual FPLs in China." *(line 12 on page 1)*

***General comment 3.*** Still related to the comparison of county level plans and the national policy regarding protection level, can you please provide some additional information on how these counties were selected? It would be relevant to understand if these counties are representative of the different realities in China, particularly in terms of the variables defined in the policy (rural/urban, exposed population, arable land). You found an agreement in FPLs in 66.7% of the counties – can this be attributed in some way to specific properties of these counties, for example? Additional information on the validation counties and additional discussion on this would be useful.

*Accepted*: Thank you for the suggestion. The validation counties are selected based on the date of the flood protection design documents: the data should represent the flood protection of the year 2015. From accessible authority websites and literature, we found a raw sample of 304 counties with flood protection documents dating from 1998 to 2019. For the first round, we only selected the relatively new documents released from 2007 to 2012, neither too old that may be outdated nor too new that may be uncompleted and unqualified. Those documents were kept in the validation data only if they stated that the design would be completed between 2010 and 2015. A sample of 110 counties was selected from this round. For a second round, new flood protection design documents starting from 2015 were researched and these were kept only if they stated the current (2015) flood protection standards. Another 61 counties were selected then. Now, we have added how the validation sample is selected in the manuscript. Please check from *lines 121–129 on page 6*.

Additionally, we clarified the representativeness of the validation sample, also following your critical General comment 2 and General Comment 3 of Referee #3. With an expanded validation sample from 51 to a total of 171 counties, the validation data include 122 urban counties (19.1%) and 49 rural counties (3.1%). These represent 34.0% of the total exposed population and 13.0% of Chinese exposed arable land. Thus, we believe the validation counties can now be taken as representative of general Chinese territory. Please check from *lines 129–131 on page 6*.

***Specific comments***

***Specific comments 1.*** Title: I feel that the use of "social divergence" raises a reader's expectations above what is actually presented in the article, which is limited to age groups. Please adjust the title to reflect this, or otherwise expand the analysis to include other factors that influence social vulnerability – the latter would certainly be more insightful and make the article more interesting.

*Accepted:* Thank you for the suggestion. We have expanded the social divergence to include the exposed rural and urban population following your critical comment. Accordingly, we revised sections 3.3 and 3.4 *(lines 219–222 on page10 and lines 240–244 on page 11)*; and added the supplementary Table S3 for the urban and rural population. However, the paper still does not consider all the aspects of social divergence, due to data limitation, which is now further clarified in section 4.3 *Limitation and future perspectives (lines 333–336 on page 15).*

***Specific comments 2.*** L37: Remove 'Each year' (I assume these are aggregate numbers for 1990-2017)

*Accepted:*  Thank you for this suggestion. It is indeed average data. We revised the sentence *(lines 39–41 on page 2)* and now it reads:

> Between 1990 and 2017, floods in China averagely affected 149 million people, led to 2165 deaths, and caused an economic damage of US$ 34 billion per year (Du et al., 2019).

*Specific comments 3.*  L58: I do not fully understand what the second research question means, in the sense that the policy document does not make reference to demographics in the definition of FPLs, and so the answer to this is already known. Please clarify.

*Accepted:*  Thank you for this suggestion. We clarified the second research question, also following General Comment 2 of Referee 3. Now it reads as: "Since the FPL policy does not consider population demographics, what are the implications for the protection of vulnerable social groups?" Please check from *lines 62–63 on page 3.*

*Specific comments 4.*  L65: My interpretation of Jonkman, 2013 is that it states the actual opposite of what you are saying in this sentence. For example, Jonkman, 2013 says that ": : : the actual protection levels could differ by more than a factor of 10 from the protection standard, and the effect on risk will be similar." Please discuss and revise.

*Accepted:*  Thanks for this suggestion. We revised the sentence *(lines 68–70 on page 3)* and now it reads:

> Flood protection policies provide an opportunity to establish a large-scale FPL dataset (Mokrech et al., 2015) as they generally contain information on how a region should be protected from floods, although some authors suggest that the actual protection levels could differ from the protection standard from policy (Jonkman, 2013).

*Specific comments 5.*  Eq. 1: I find "GDP-weighted PopE" a poor name for a variable, as it is a bit long and at first sight it appears to be GDP minus: : : Please improve.

*Accepted:*  Thanks for this suggestion. We also would prefer a shorter variable name, but this would require another abbreviation, while we think we have enough. We have changed it to "GDP weighted PopE".

*Specific comments 6.*  L119: Section 2.5 is unexpected and feels disconnected from what comes before in the article, because up to this point you have not yet stated that this is an analysis you will be doing. Is this cluster analysis meant to address a research question? Please contextualize beforehand, and when doing so provide an explanation on why this analysis is useful.

*Clarified:*  Thank you for the suggestion. This section is associated with the first research question "What level of protection against river floods does Chinese policy imply across the country?" Based on the derived flood protection levels (FPLs), we can have a map and describe the distribution of the FPLs (high values and low values). More than that, the spatial pattern analysis of the FPL data quantitatively shows where the significant high/low values are located and how the high/low values are clustered. We think this method is important, as it adds a rigorous spatial analysis. Meanwhile, it can present the regional risk: a high-FPL county should

also be at risk if its surroundings suffer severe flooding. We added an explanation of this at *lines 135–138 on page 6*.

***Specific comments 7.*** L163: Unclear which previous studies this sentence refers to. Is it only Scussolini et al., 2016? Please clarify.

*Accepted:* Thank you for the suggestion. This sentence has been clarified as follow *(lines 182–183 on page 8)*:

> Therefore, Chinese FPLs are probably underestimated in previous studies (Scussolini et al., 2016).

***Specific comments 8.*** L197: Because FPLs also change over time but only current FPLs are considered in this section, I am unsure about the usefulness of the analysis carried out here. For the same reason, I also find this section title a bit misleading. Please improve and clarify.

*Clarified:* Thank you for the suggestion. Indeed, both FPLs and population change over time. In this section, we focus on how the exposed population changes if flood protection is kept constant over time. Such a method clearly and directly shows how the total population and the demographic characteristics changed in areas of currently different flood protection levels and how the change rate varied between current high and low flood protection levels. We believe such a strategy can clearly indicate the importance of considering population dynamics and demographic characteristics in the flood protection policy, which is critical for improving the policy.

***Specific comments 9.*** L235: This could also simply be the result of FPLs being calculated on the basis of present-time exposed population, couldn't it? We do not have information about FPLs in 1990; therefore, stating that a faster increase in exposed population may have occurred in these counties because in the past their FPL was already high seems speculative. Please discuss.

*Accepted:* Thanks for the suggestion. We have revised this sentence to avoid confusion. Now the sentence reads as follows *(lines 265–267 on page 12)*:

> The possibility of a similar outcome should be considered in China, as suggested by the faster increasing trend of the exposed population in the high-FPL counties than in the low-FPL counties.

***Specific comments 10.***   Table 1: Note at the bottom is unclear.

***Accepted:***   Thanks for the suggestion of Table 1. This note has been clarified. Please check from *lines 514–518 on page 21,* or as below.

**Table 1.** Urban and rural standards for evaluating the flood protection level (FPL) (source: Standard for flood control GB 50201-2014)

| Urban FPL Indicators | | Urban FPL (Return period, years) | Rural FPL Indicators | | Rural FPL (Return period, years) |
|---|---|---|---|---|---|
| Population exposure (million) | GDP weighted population exposure* (million) | | Population exposure (million) | Arable lands exposure (thousand ha) | |
| <0.2 | <0.4 | 30–50 | <0.2 | <20 | 10–20 |
| ≥0.2 | ≥0.4 | 50–100 | ≥0.2 | ≥20 | 20–30 |
| ≥0.5 | ≥1 | 100–200 | ≥0.5 | ≥66.7 | 30–50 |
| ≥1.5 | ≥3 | ≥200 | ≥1.5 | ≥200 | 50–100 |

Note: * GDP weighted population exposure is the population exposure multiplied by the ratio between the relative per capita gross domestic product (GDP) and the national average.

***Specific comments 11.***   Figure 2: Remove "the" in y-axis label.

***Accepted:***   Thanks for the suggestion for Figure 2. This label has been corrected. Please check from *lines 526–527 on page 24,* or as below.

[Figure]

**Figure 1** The number of counties with different flood protection levels. (The map of western and eastern China is shown in Figure 3)

***Specific comments 12.***   Figure 5: In the y-axis label, replace "Exposure" with "Exposed population".

***Accepted:***   Thanks for the suggestion. This label has been corrected. Please check from *lines 534–535 on page 27,* or as below.

[Figure]

**Figure 5** Changes in exposed total population **(a)** and vulnerable population **(b)** across different flood protection levels from 1990 to 2015.

---

## Author Comment (AC3) · 30 Nov 2020

**Response to the Anonymous Referee #3**

***General Comments.*** The objective of the paper is to develop and validates a Flood Protection Level (FPL) dataset for China, which is based on current Chinese policy on FPLs. Accordingly, base data and methodologies for its development are first discussed, and then results are critically analysed. Although the paper does not represent any significant improvement in research, it supplies relevant information for flood risk management in one of the biggest and most flood prone area of the world; and thus, it can be of interest for the journal audience. The paper is generally well written and organised; data, methods and results are quite well explained. However, before results can be published, shared and made available to the research community, I think that some conceptual aspects deserve more attention and clarification.

*Accepted*: Thanks for confirming the relevance of our manuscript and the suggestions for further improvement. We have thoroughly revised the paper, addressing your valuable comments and suggestions.

***General Comment 1.*** The FPL generated by the dataset is a theoretical one (i.e. designed based) and not the real one. This must be very clear since the beginning of the paper and not marginally discussed at the end. Accordingly, authors should stress since the beginning why this information is useful, how it can be used for risk management, e.g. as a proxy of the flood risk in an area?

*Clarify*: Thank you for the suggestion. We have revised the paper accordingly in the *Introduction*. First, we further provide arguments for why the data is useful. Nowadays, flood risk assessment is drawing an increasing attention worldwide and playing a critical role in flood risk management. However, flood protection information, an essential element of flood risk estimates, is rarely available in reality, which dampens a reliable analysis of flood risk and its applications. Particularly for China, the only nation-wide available data is from the global database FLOPROS (Scussolini et al., 2016), which has a raw resolution of provinces. On the other hand, the Chinese flood control policy clearly stated how an area should be protected according to the exposed elements. We believe this information is useful for risk analysis and management. Please check from *lines 20–58 on page 1–3*.

Second, we added how the newly developed database can be used for risk management, also following your Minor Comment 9. 1) Authorities can use this database to check if the relevant counties are protected properly. 2) Flood risk assessment could be conducted considering the developed flood protections. 3) The policy-based FPL can be an important foundation for relevant researchers to develop a more reliable FPL dataset of China and for the rest of the world. 4) It can help to reveal potential social divergence by combining the policy-based FPL with demographic data, which can further improve the flood protection policy, as indicated by the relevant analyses in this study. We have discussed this issue in Section 4.3. Please check from *lines 316–321 on page 14*.

***General Comment 2.*** With respect to the last point, the second research question could then be changed in: Is FPL representative of the real risk in the area or its definition/evaluation should be changed? In fact, the present second research question (i.e. does the FPL policy take into account relevant demographics of the exposed population, such as elders and children who are known to be most vulnerable to floods?) is not clear at this point of the paper (i.e. why exactly this question?) as it is too much linked to an evidence that comes out only at the end of the manuscript

***Accepted***: Thank you for the suggestion. Also following Specific Comment 3 of Referee #2, we have revised the second research question to "Since the FPL policy does not consider population demographics, what are the implications for the protection of vulnerable social groups?" Please check from *lines 62–63 on page 3*.

***General Comment 3.*** The validation process is very weak, so I do not agree with authors that theoretical FPL agrees with real one very well (see section 4.3). The validation process was carried out only for 51 (about 2%) out of 2237 counties and a match was observed only for 66.7% of the counties (abut 1,5%). This has important implication on the use of results (see comment 1)

***Accepted***: Thank you for the critical suggestion. We have made the following two efforts to strengthen the validation, also considering General Comment 2 of Referee #2.

First, we increased the validation sample size from 51 counties to 171 counties. Now, the match ratio between the FPL database and the validation data is 53.2%. It can reach 90.1% if we apply a free bound of one protection level (the protection levels are considered as a match if the difference is zero or one protection level). Please check from *lines 170–175 on page 8*, and *Supplementary* Table S1 and Table S2.

Furthermore, we also discussed the representativeness of the validation sample. It represented 34.0% of the total exposed population and 13.0% of exposed arable lands in China. Thus, we believe the expanded samples should provide a relatively more reliable validation. Please check from *lines 129–131 on page 6*.

***General Comment 4.*** The calculation of FPL is based on the assumption that the exposed area coincides with the 100 years return period flooded area. As this critically affects the estimation of FPL, authors should explain the reasons of this assumption. Moreover, how such an area was derived? does the modelling consider or not the existence of flood protections? What this implies?

***Accepted***: Thank you for the suggestion. We clarified the flood exposure definition and the employed flood data, also following General Comment 1 of Referee #2.
First, we calculated the flood exposure as the elements within the maximum extent of the 100-year return period flood. This definition is consistent with the flood risk assessment by Shi et al (2015) and the flood exposure analysis by Jongman et al (2012), Du et al (2018), and Fang et al (2018). Please check from *lines 104–107 on page 5*.

Second, the 100-year flood map we applied is undefended, which was provided by Dr. Roberto Rudari from the CIMA Foundation. This dataset was produced based on hydrological and hydraulic models at a resolution of 1 km, which were validated against historical floods. And it has been effectively used for analyzing China's urban land expansion (Du et al., 2018) and population dynamics in floodplains (Fang et al., 2018). The undefended data were used instead of the defended one for two major reasons. First, the flood defenses were designed based on the protected population and assets, as shown in the Chinese flood control policy. Second, flood defenses cannot ensure the protected areas' absolute safety; thus, the population and assets should not be excluded from flood exposure analysis. We now specify this important feature in the manuscript. Please check from *lines 88–91 on page 4*.

*References:*

*Du S, He C, Huang Q, Shi P, 2018. How did the urban land in floodplains distribute and expand in China from 1992–2015? Environmental Research Letters, 13(3): 034018.*

*Fang Y, Du S, Scussolini P, Wen J, He C, Huang Q, et al., 2018. Rapid Population Growth in Chinese Floodplains from 1990 to 2015. International Journal of Environmental Research and Public Health, 15(8): 1602.*

*Jongman B, Ward P J and Aerts J C J H 2012 Global exposure to river and coastal flooding: long term trends and changes Glob. Environ. Change 22 823–35*

*Shi PJ, Wang JA, Xu W, Ye T, Yang SN, Liu LY, Fang WH, Liu K, Li N and Wang M. 2015 World Atlas of Natural Disaster Risk (Heidelberg: Springer)*

*Minor comments*

*Minor comments 1.*    line 21 "With the emergence of large-scale flood models, the necessity to quantify FPLs has increased in recent years" the cause-effect relation is not clear to me, could authors comment more on this?

*Accepted*:    Thanks for your suggestion. We have revised this sentence accordingly (*lines 22–23 on page 1*). Now it reads as:

> With increasing focus on large-scale flood risk assessment, which also depends critically on flood protection information (Ward et al., 2017; Alfieri et al., 2017; Winsemius et al., 2018), the necessity of quantifying FPLs has increased in recent years.

*Minor comments 2.*    line 27 what "improved FPLs" means?

*Accepted:*    Thank you for this suggestion. It means high FPLs. We revised the sentence (*line 29 on page 2*) and now it reads:

> High FPLs reduce the frequency of floods in flood-prone areas and decrease flood risk (Ward et al., 2013).

*Minor comments 3.* line 37 "China is one of the countries that experience the most serious floods and the fastest urbanization. Each year between 1990 and 2017, floods in China affected 149 million people, led to 2165 deaths, and caused an economic damage of US$ 34 billion" I guess these figures refer to average data

*Accepted:* Thank you for this suggestion. It is indeed average data. We revised the sentence (*lines 39–41 on page 2*) and now it reads:

> Between 1990 and 2017, floods in China averagely affected 149 million people, led to 2165 deaths, and caused an economic damage of US$ 34 billion per year (Du et al., 2019).

*Minor comments 4.* line 85 "It originally has a spatial resolution of 100 m and is aggregated to a 1 km resolution to match the flood depth data, further to get population exposure using methods described in Fang et al. (2018)" I think that a brief explanation/recall of how the data were elaborated is required.

*Accepted:* Thanks for the suggestion. A brief explanation and relevant references were added. Please check from *lines 104–107 on page 5*.

*Minor comments 5.* line 151 "In 34 (66.7%) out of the 51 verification counties, the FPLs agree with the local official protection plans (full information in Supplement). The FPLs in the dataset are overestimated in four counties and underestimated in five counties" what about the other 8 counties?

*Accepted*: Thank you for the suggestion. It was a mistake. We revised the sentence with an expanded validation sample *(lines 169–172 on page 8)*. The sentence now reads as follow:

> In 91 (53.2%) out of the 171 verification counties, the FPLs agree with the local official protection design documents (Supplementary Table S1 and S2). The FPLs in the dataset are overestimated in 20 counties (11.7%) and underestimated in 60 counties (35.1%).

*Minor comments 6.* line 176 "These counties within the "low-high" FPL clusters can be more vulnerable when they are needed to sacrifice to protect their surrounding large cities that are more expensive to be flooded" not clear, more vulnerable than what? Could authors explain?

*Accepted:* Thank you for this suggestion. We have revised the sentence *(lines 196–199 on page 9)* and it reads as follows:

> These counties within the "low-high" FPL clusters can be vulnerable to floods when they are needed to sacrifice to protect their surrounding large cities that are more expensive to be flooded (Wang et al., 2016). For instance, in China, flood detention zones are planned in rural areas to protect surrounding cities in the Yangtze River and Huaihe River Basins of China (Du et al., 2020).

*References:*

*Du, S., Shen, J., Fang, J., Fang, J., Liu, W., Wen, J., Huang, X., and Chen, S.: Policy delivery gaps in the land-based flood risk management in China: A wider partnership is needed, Environmental Science & Policy, 116, 128-135, https://doi.org/10.1016/j.envsci.2020.11.005, 2021.*

*Minor comments 7.* line 217 "The newly developed data show that almost one third (33.1%, 741) of the evaluated Chinese counties are protected with a ⩾30-year FPL" should be protected…. It's a theoretical FPL

*Accepted*: Yes, it's a theoretical FPL. Accordingly, this sentence has been revised as follows:

> The newly developed data show that almost one third (33.1%, 741) of the evaluated Chinese counties are should be protected with a ≥30-year FPL, while this FPL is only in 5 (14.7%) out of 34 provinces in the FLOPROS (Scussolini et al., 2016). *(lines 248–250 on page 11)*

***Minor comments 8.*** line 224 "For instance, global flood risk assessments show huge flood risk across Chinese provinces both in current condition and future scenarios (Willner et al., 2018a), which are considered to further propagate a devastating indirect impact to other countries through the global trade and supply network (Willner et al., 2018b). However, those global assessments are based on the FLOPROS database, which significantly underestimate Chinese FPLs, e.g., presenting Beijing with a 20-year FPL, which should be 200 years in the newly developed result (Fig. 3) and in the local official document (full information in Supplement). The real flood risk should thus be much lower than the estimates in previously studies if the new FPL is considered" The authors cannot made this statement as the correspondence between theoretical and real FPLs have been evaluated only for 51 out of 2237 counties; the case of Beijing is a fortunate one where a perfect match occurs. But, can authors exclude that counties exist where there is not a FPL at all in practice, in front of a theoretical FPL, or a real FPL that is lower to designed based one? In this case, the risk can be underestimated. Please, comment.

*Accepted*: Thanks for the suggestion. We have increased our validation sample size from 51 to 171, also following your critical General Comment 3 and General Comment 2 of Referee #2. And now the validation samples represent 7.6% of the surveyed Chinese counties, 34.0% of the exposed population, and 13.0% of Chinese exposed arable land. Besides, the FPL dataset has a higher resolution than the FLOPROS; the former is based on counties and the latter is based on provinces. Therefore, we believe the FPL data are a valuable proxy. Please check from *lines 129–131 on page 6.*

Meanwhile, the reason for the overestimation of Chinese flood risk mainly results from an underestimation of Chinese protection against flood. Therefore, we have compared the difference between FPL and FLOPROS by Paired Sample T Test. Please check from *Supplementary* Table S4. Furthermore, we have revised the sentence *(lines 257–259 on pages 11–12)* and it reads as follows:

> However, those global assessments are based on the FLOPROS database, which is significantly lower than the policy required FPLs as indicated by the Paired Sample T Test (p<0.01, supplementary Table S4). For instance, FLOPROS presented Beijing with a 20-year FPL, while it should be 200 years according to the Chinese protection policy (supplementary Table S1).

***Minor comments 9.*** line 245 "A neglect of the real-world flood protection lagging behind the policy-based flood protection can distort the selection of adaptation measures" this is exactly the point. Then, how theoretical FPL can be used (see general comment 1)?

*Accepted*: Thanks for the suggestion. Also following your General Comment 1, we added a discussion on how the theoretical FPL can be used:

> 1) The authorities can use this database to check if the relevant counties are

protected properly. 2) Flood risk assessment could be conducted considering the developed flood protections. 3) The policy-based FPL can be an important foundation for relevant researchers to develop a more reliable FPL dataset of China and the rest of the world. 4) It can help to reveal potential social divergence by combining the policy-based FPL with some social data, which can further improve the flood protection policy, as indicated by the relevant analyses in this study.

***Minor comments 10.*** line 266 "Such a strategy, however, may aggregate flood risk because the less protected areas coincide with high social vulnerability that is caused by a disproportional distribution of vulnerable people, particularly elders" what authors mean with "aggregate flood risk"

*Accepted*: Thanks for your suggestion. It was unfortunately a misspelling. It should be "aggravate", which was corrected (*lines 299–301 on page 13*).

***Minor comments 11.*** line 300 "This study thus agrees with the argument of Scussolini et al. (2016) that flood protection policy is a valid proxy for actual FPL" I do not agree, see general comment 4

*Accepted*: Thanks for the suggestion. We revised the paper accordingly. First, we expanded the validation sample size from 51 counties to 171 counties, also following your insightful General Comment 3. Second, we revised the statement as follows:

This study thus agrees with the argument of Scussolini et al. (2016) that flood protection policy is a valuable proxy for actual FPLs. *(lines 341–342 on page 15)*

***Minor comments 12.*** Figure 1 I think that a full description of the framework is required in the text, i.e. in Section 2.1, to support readers in the full comprehension of following contents.

*Accepted:* Thanks for your suggestion. A full description of the framework has been added. Please check from *lines 78–84 on page 4*.

***Minor comments 13.*** Figure 3 colours used for the FPLs 30-50 and 50-100 cannot be distinguished in the figure.

*Accepted:* Thanks for the suggestion. *Figure 3* has been improved accordingly. Please check from *lines 528–531 on page 25* or as below.

[Figure]

**Figure 3** Flood protection level (FPL) for Chinese counties. The FPL is limited to the scope of floodplains but plotted to cover the entire counties. The data should only be viewed as a proxy of the actual FPLs, not equating to the actual FPLs. The Shapefile format data are available as a supplement.

---

## Referee Report (RR1)

Journal: NHESS
Title: **Assessing Chinese flood protection and its social divergence**
Author(s): Wang et al.
MS No.: NHESS-2020-264
MS Type: Research Article
**Iteration: Second review**

I really thank the authors for the efforts made in the revision of the paper. Integrations made by authors addressed and clarified most of my comments. However, my main objection on the validation process still remains. Despite the fact that the validation dataset has increased, correspondence between theoretical and real FPLs has been proved only for 91 (4%) out of 2237 counties. I totally agree that, for many aspects, the present database constitutes a substantial improvement of previous knowledge of Chinese FPLs, and also that the present database is very useful for flood risk analysis and for an improved flood risk management in China, but I do not agree that present database reflects real FPLs. I would then encourage authors to be less confident in statements like:

"The policy-based database is a valuable proxy for actual FPLs in China"

"Chinese FPLs are significant underestimated in previous studies"

"It does not report actual FPLs although it agrees with local flood protection design documents very well"

---

## Author Response (AR2)

**Point-by-point response to editor's and reviewers' comments**

**Part 1: Author's Responses to Comments from Editor Heidi Kreibich**

Comments made by the editor are shown in black text.

Author responses are provided in blue text.

**Comments:** There remain just a few minor aspects which still need to be resolved, see comments of the two reviewers. Particularly, you need to be more precise, honest and maybe critical about what you can provide (and what you are not able to provide) with your study.

**Accepted:** Thanks for confirming the quality of our paper and the first-round revision, and providing valuable suggestions and comments for further improvement. We have thoroughly checked the paper to be more precise, honest, and critical about our findings, particularly about the use of the policy-based FPLs.

**Part 2: Author's Responses to Referee Report #1 (Comments from the Anonymous Referee #3)**

Comments from Referee Report #1 are provided in black text.

Author responses are provided in blue text (line and page numbers refer to the clean version).

**General Comments.** I really thank the authors for the efforts made in th3e revision of the paper. Integrations made by authors addressed and clarified most of my comments. However, my main objection on the validation process still remains. Despite the fact that the validation dataset has increased, correspondence between theoretical and real FPLs has been proved only for 91 (4%) out of 2237 counties. I totally agree that, for many aspects, the present database constitutes a substantial improvement of previous knowledge of Chinese FPLs, and also that the present database is very useful for flood risk analysis and for an improved flood risk management in China, but I do not agree that present database reflects real FPLs.

**Accepted:** Thanks for confirming the quality of our manuscript and the significance of the developed database. We have further revised the paper according to your valuable comments and suggestions. Particularly, we clearly stated the difference between the policy-based database and the real FPLs.

**Detailed Comment.** I would then encourage authors to be less confident in statements like:

"The policy-based database is a valuable proxy for actual FPLs in China"

"Chinese FPLs are significant underestimated in previous studies"

"It does not report actual FPLs although it agrees with local flood protection design documents very well"

**Accepted:** Thank you for the suggestion. The three sentences have been clarified as follows:

- This suggests that the policy-based FPLs could be a valuable proxy for designed FPLs in China. *(lines 11–12 on page 1)*
- Therefore, Chinese FPLs are probably underestimated in the previous study by Scussolini et al. (2016). *(lines 183–184 on page 8)*
- It does not report actual FPLs although it generally agrees with local flood protection design documents. *(line 314 on page 14)*

**Part 3: Author's Responses to Referee Report #2 (Comments from the Anonymous Referee #2)**

Comments from Referee Report #2 are shown in black text.

Author responses are provided in blue text (line and page numbers refer to the clean version).

***General Comments.*** The authors have addressed most of the identified issues and have significantly improved the manuscript in this revision. In my opinion, the article is mostly ready for publication, after the following issues are addressed.

*Accepted:* Thanks for confirming the quality and significance of our manuscript. We have revised the paper, addressing your valuable comments and suggestions.

***Detailed Comments 1*** *(referring to General comment 2 in Revision Round #1).* The authors have expanded their validation dataset, which is undoubtedly positive. Nevertheless, this is still a validation against design documents, not actual protections. I continue to believe that this limitation must be more clearly acknowledged from the start. For example, the abstract reads: "The new dataset corresponds with local flood protection designs in 91 (53.2%) of the 171 validation counties, and in 154 counties (90.1%) it is very close to the designed FPLs. This suggests that the policy-based FPLs is a valuable proxy for actual FPLs in China." But the paper provides no evidence that policy-based FPLs are a good proxy for actual FPLs, so this remains a supposition. The conjecture provided in the reply is equally unsupported (i.e. "we believe it is plausible to assume the actual protection of protection infrastructures that are completed and qualified to be equal to or higher than the designed standards, as a result of strict and tight control in the authoritarian administration of China."). Please refer to your own reply to specific comment 4.

*Accepted:* Thank you for confirming the significance of our validation and providing valuable suggestions. We have further clarified the limitation of the policy-based FPLs throughout the manuscript. The sentences you mentioned were clarified as follows:

> "The new dataset corresponds with local flood protection designs in 91 (53.2%) of the 171 validation counties, and in 154 counties (90.1%) it is very close to the designed FPLs. This suggests that the policy-based FPLs could be a valuable proxy for designed FPLs in China." *(lines 10–12 on page 1)*

***Detailed Comments 2*** *(referring to Specific comment 7 in Revision Round #1).* It is still not clear if this refers to one single study, or to multiple studies: you use plural but have only provided one reference. Please clarify.

*Accepted:* Thank you for the suggestion. This sentence has been clarified as follow:

> Therefore, Chinese FPLs are probably underestimated in the previous study by Scussolini et al. (2016). *(lines 183–184 on page 8)*

***Detailed Comments 3*** *(referring to Specific comment 8 in Revision Round #1).* I do not think this reply adequately addresses my comment and I remain unsure about the usefulness of this analysis, as the evolution of FPLs is not analysed together with the evolution of the population. The authors should at least acknowledge this limitation in the article.

***Accepted:***    Thanks for the important suggestion. We have added the limitation in the section *4.3 Limitations and future perspectives* and now it reads:

[revised manuscript text omitted]